# Phylogenetic Reconstruction of the Rainforest Lineage *Fontainea* Heckel (Euphorbiaceae) Based on Chloroplast DNA Sequences and Reduced-Representation SNP Markers

**Aaron J. Brunton** [1,2,*] , **Robert W. Lamont** [1,2] , **Gabriel C. Conroy** [1,2] , **Samantha Yap** [3] , **Maurizio Rossetto** [3] , **Alyce Taylor-Brown** [1,4] , **Laurent Maggia** [5,6,7] , **Paul W. Reddell** [8] and **Steven M. Ogbourne** [1,2]

1   Centre for Bioinnovation, University of the Sunshine Coast, Maroochydore, QLD 4556, Australia
2   School of Science, Technology and Engineering, University of the Sunshine Coast, Maroochydore, QLD 4556, Australia
3   Research Centre for Ecosystem Resilience, Australian Institute of Botanical Science, Royal Botanic Garden Sydney, Sydney, NSW 2000, Australia
4   Parasites & Microbes Programme, Wellcome Sanger Institute, Wellcome Genome Campus, Hinxton, Cambridge CB10 1SA, UK
5   Institute for Exact and Applied Sciences, Université de la Nouvelle-Calédonie, 98851 Nouméa, New Caledonia
6   Centre de Coopération Internationale en Recherche Agronomique pour le Développement (CIRAD), UMR AGAP, 98800 Nouméa, New Caledonia
7   AGAP, Institut Agro, CIRAD, INRAE, University Montpellier, 34070 Montpellier, France
8   EcoBiotics Ltd., Yungaburra, QLD 4884, Australia
*   Correspondence: aaron.brunton@research.usc.edu.au

**Abstract:** *Fontainea* is a plant genus with nine recognised species that occur across the tropical and subtropical rainforests of Australia, Papua New Guinea, New Caledonia, and Vanuatu. One of these species is cultivated commercially as the source of a cancer therapeutic, and several other species are under threat of extinction. Despite this, the phylogenetic relationships of the genus have not been explored. Our study assessed the phylogeny of seven *Fontainea* taxa from the Australian and Pacific Island complex using chloroplast DNA sequence data and reduced-representation genome sequencing. Maximum-likelihood and consensus network trees were used to infer the topology of phylogenetic relationships between species, which highlighted three distinct lineages and a number of sister species. Our results indicated that the geographically disjunct species *Fontainea venosa* and *F. pancheri* formed a sister group at the earliest position of divergence for the genus. The data also revealed that the vulnerable *Fontainea australis* and the critically endangered *F. oraria* form a sister subclade with evidence of some shared plastid genotypes. Generally, our phylogenetic reconstruction supports the modern taxonomical nomenclature. However, we suggest further accessions across several species may support improved genetic distinctions between the sister groups of *Fontainea* within the genus.

**Keywords:** molecular divergence; phylogeography; single nucleotide polymorphisms; oncology; natural product; plastids; rainforest evolution

## 1. Introduction

*Fontainea* Heckel is a genus of dioecious rainforest shrubs or small trees classified in family Euphorbiaceae, subfamily Crotonoideae, tribe Codiaeae [1], with ancestral links to the Tertiary period [2]. Currently, there are nine recognised species that occur in Australia, Papua New Guinea, New Caledonia, and Vanuatu, often in fragmented populations [3–5]. Five of the Australian species are classified as threatened, while the two Papua New Guinea species are poorly studied with limited herbarium collections. The sole Australian species not classified as threatened, *Fontainea picrosperma* C.T. White, is of scientific and commercial interest following the discovery of a novel anti-cancer agent, tigilanol tiglate, from the

plant [6]. Tigilanol tiglate has subsequently been approved for use in the USA, Europe, and Australia as a veterinary pharmaceutical for treating canine mast cell tumours [7]. As such, the genus *Fontainea* provides an example of the conservation value of plant biodiversity not only for ecological and social values but also as a potential source of genetic material for future medicines and other valuable products.

There are limited fossil records from the Euphorbiaceae family in Australia. However, phylogenetic analysis using sequence data showed an Australian origin of *Fontainea* [8,9]. *Fontainea* are part of a subclade of a mostly Australian clade, which also contains the genera *Baloghia*, *Beyeria*, and *Ricinocarpus* [8,10]. Currently, there are no fossil records of extant *Fontainea*; yet, Australian macrofossil evidence of silicified fruit suggest an Oligocene-Miocene origin for the closely related *Fontainocarpa* [11]. It is conceivable that *Fontainocarpa* represents a *Fontainea* fossil fruit even though its position in the lineage is presently uncertain and open to debate position.

Current taxonomical classifications of *Fontainea* are based on traditional morphological features, with the nine recognised species being *F. australis* Jessup & Guymer, *F. borealis* P.I. Forst., *F. fugax* P.I. Forst., *F. oraria* Jessup & Guymer, *F. pancheri* (Baill.) Heckel, *F. picrosperma*, *F. rostrata* Jessup & Guymer,*F. subpapuana* P.I. Forst, and *F. venosa* Jessup & Guymer, which display, for instance, the difference in stamen number, width and shape of interstitial faces of endocarp, presence of a swollen leaf petiole, and position of basilaminar glands—see Jessup and Guymer, [3], and Forster, [4], for the complete detailed morphological descriptions and key to the taxa. Chloroplast DNA (cpDNA) sequence markers have been applied to numerous phylogenetic studies due to their highly conserved nature and generally low recombination rate of sequences [12]. However, genome-wide data have become more accessible and utilised alongside cpDNA markers to better resolve the phylogeny of closely related species [13,14].

With the emergence of new molecular tools for assessing phylogenetic relationships, it is valuable to revisit the current traditional classifications, especially to ensure effective conservation management of threatened species. Recently, there have been several species-specific population genetic and floral biology studies, of the Australian *Fontainea* species [15–19], with one study [17] indicating the species boundaries of two threatened taxa (*F. australis* and *F. oraria*) need further attention. Rossetto [17] highlighted some populations of *F. australis* were genetically different from other populations and did not cluster with morphologically similar *F. oraria*. Clearly, a molecular-based study that examines relationships within and between *Fontainea* species is needed to better manage conservation efforts for this commercially and ecologically significant rainforest genus.

Recent progress in next-generation sequencing (NGS) technology provides access to low-cost, rapid, and accurate high-throughput data on a genome-wide scale. An example of a reduced-representation genome sequencing platform is Diversity Arrays Technology sequencing (DArTseq). DArTseq is used in combination with NGS to generate large volumes of single nucleotide polymorphism (SNP) markers without the need for a reference genome of the target species. This method utilises a complexity reduction technique to produce a genome representation, which is then used for genotyping individuals. There is a growing body of research in which DArTseq has been applied to accurately resolve phylogenetic relationships for a variety of plant species, including wild and commercial populations [13,20,21]. As such, in the context of providing an accurate estimate of the phylogeny for this genus of high conservation significance, there is an opportunity to employ NGS tools in conjunction with more traditional phylogenetic tools (cpDNA sequencing) to better inform species management and evolutionary scenarios.

In this study, we used plastid DNA (cpDNA) sequence markers and DArTseq data coupled with sequences extracted from GenBank to examine the phylogenetic relationships of a complex group of *Fontainea* (Austral-Pacific). In addition, we compared outcomes from the different marker types and present the first phylogeny focussed on a broad group of *Fontainea* species, which allows for a comparison between the modern systematic treatments.

## 2. Materials and Methods

*Fontainea* occur across environmentally heterogeneous regions of the tropical and subtropical biomes of Australia, PNG, New Caledonia, and Vanuatu (Figure 1). A range of rainforest habitats are represented across the distribution of the genus, including wet tropical rainforest (*F. borealis*, *F. picrosperma*, and *F. subpapuana*), seasonally dry rainforest (*F. fugax*, *F. pancheri*, *F. rostrata*, and *F. venosa*), aseasonally wet subtropical rainforest (*F. australis*), and littoral rainforest (*F. oraria*). In Australia and PNG, all species typically appear on red-brown, basalt-derived, krasnozem soils [3]. New Caledonia has a complex and widely contended geological history [22,23]; nonetheless, *F. pancheri* on Grande Terre and the Loyalty Island of Lifou are typically found on more or less deep or stony soil on limestone and sedimentary substrates.

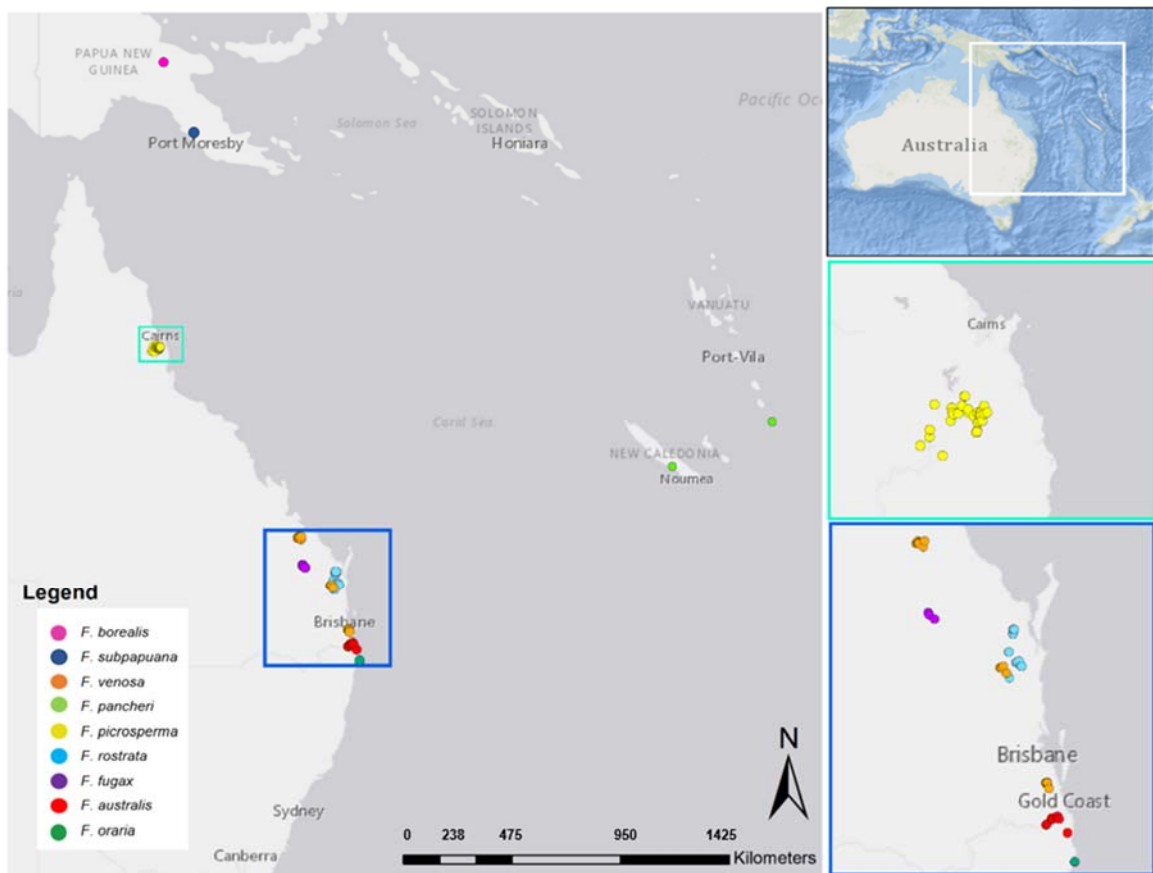

**Figure 1.** Geographic distribution of the *Fontainea* genus. Coloured boxes highlight the current known distribution of Australian *Fontainea* species.

### 2.1. Study System and Sampling Strategy

A total of 59 accessions from 7 *Fontainea* species and two outgroup taxa were analysed by data generated from cpDNA and DArTseq SNP markers, combined with published sequence data obtained from GenBank (Table S1). Leaf material was collected from 57 accessions from mature individuals of all recognised *Fontainea* species (Table 1) except for the PNG species (*F. borealis*, *F. subpapuana*). Following a similar sampling strategy of Rossetto, Bragg [24], we selected a minimum of six samples per *Fontainea* species to provide an adequate genomic representation of the intraspecific variation at the species level for the DArTseq SNP data. Noting the fragmented and isolated nature of *Fontainea*, we aimed to include accessions from across the geographical range of each respective species. *Fontainea venosa* has the broadest distribution of the genera, generally occurring in three separate regions: Boyne Valley, Gympie, and Bahrs Scrub, QLD. There is a distance of ~550 km

between the northern (Boyne Valley) and southern (Bahrs Scrub) regions. However, we only had *F. venosa* samples available to be genotyped from the Boyne Valley and Gympie regions. *Fontainea australis* accessions are from the original type (Limpinwood) described by Jessup and Guymer [3] and three additional types across the northern and eastern range of the species. Sampling for *F. fugax*, *F. rostrata*, and *F. picrosperma* captured accessions from most known locations of each species. *Fontainea oraria* accessions are wild types from the only known natural population at Lennox Head, NSW. Where possible, we used matching accessions for the cpDNA and DArTseq datasets (Table S1).

**Table 1.** List of *Fontainea* species, geographical distribution, and sampling information used for cpDNA and reduced-representation SNP analyses.

| Species | Conservation Status | Accession Distribution | N | Biome | Forest Type | Elevation (Above Sea Level) |
|---|---|---|---|---|---|---|
| *F. pancheri* | LC–1 | New Caledonia, Gladstone, and | 6 | Subtropical | Seasonally dry | ~100–400 m |
| *F. venosa* | V–2 | Gympie, QLD, Australia | 9 | Subtropical | Seasonally dry | ~150–350 m |
| *F. picrosperma* | LC–2 | Atherton-Evelyn Tablelands, QLD, Australia | 9 | Tropical | Upland wet tropical | ~700–1100 m |
| *F. rostrata* | V–2, 3 | Gympie and Maryborough, QLD, Australia | 9 | Subtropical | Seasonally dry | ~150–350 m |
| *F. fugax* | E–2 | Gayndah, QLD, Australia | 6 | Subtropical | Seasonally dry | ~350–400 m |
| *F. australis* | E–2, 3, 4 | Tweed Caldera, QLD and NSW, Australia | 9 | Subtropical | Aseasonally wet subtropical | ~50–900 m |
| *F. oraria* | CE–3, 4 | Lennox Head, NSW, Australia | 9 | Subtropical | Littoral (Coastal) rainforest | ~60 m |

LC, least concern; V, vulnerable; E, endangered; CE, critically endangered; 1, IUCN; 2, QLD Nature Conservation Act 1992; 3, Australia Wide: EPBC Act; 4, NSW Threatened Species Act 1995.

An accession from a Gympie type of the closely related taxa, *Baloghia inophylla* (G. Forst.) P.S. Green [8], was included as an outgroup. We expected this outgroup to provide suitable ancestral context for the study species, as it occurs in similar habitats and is widely distributed in both eastern Australia and some Pacific Islands, including New Caledonia and Vanuatu. Tokuoka [25] showed *Fontainea* as a sister member in a clade of *Baloghia*, *Cocconerion*, and *Ricinocarpos*. This same relationship was found in Sun, Naeem [26], while Wurdack, Hoffmann [5] showed *Fontainea* as sister to *Baloghia*. Given the contrast in these reported relationships, we also included an additional outgroup representative (*Aleurites moluccana*) external to the *Fontainea-Baloghia-Cocconerion-Ricinocarpos* clade. Although including outgroup representatives of *Cocconerion* and *Ricinocarpus* from within the Crotonoideae could be informative, these are poorly characterized with few gene regions available and are not optimised for the *Fontainea* DArTseq platform and thus were deemed inappropriate to use with *Fontainea* and *Baloghia* cpDNA and DArTseq sequence alignments. Further, the position of *Cocconerion-Ricinocarpos* among the *Fontainea* tribe appears well-resolved [10,26].

Therefore, the NCBI database (www.ncbi.nlm.nih.gov, accessed 23 March, 2022) was used to obtain sequences from an appropriate external outgroup from an additional Euphorbiaceae subfamily representative, which could be appropriately aligned with the cpDNA regions and DArTseq SNPs. We selected *Aleurties moluccana* (L.) Willd for its external position to the *Fontainea-Baloghia-Cocconerion-Ricinocarpos* clade and the broader Ricinocarpeae s.l. [10].

### 2.2. Chloroplast Sequence Data

Silica-dried leaves were used for DNA extraction and purification using a DNeasy Plant mini Kit (Qiagen Inc., Valencia, CA, USA). Plastid markers from the regions *trnQ–rps16*, *psbA–trnH*, and *petD intron D4* were selected to generate cpDNA sequence data. These plastid markers were among a set of seven regions informative at detecting variation between identified Australian plant lineages by Byrne and Hankinson [27]. For *psbA-trnH* primers, we used the forward (*psbA*F-GTTATGCATGAACGTAATGCTC) [28] and reverse (*trnH*R-CGCGCATGGTGGATTCACAAATC) [29], which has demonstrated specific promise across several phylogenetic studies of Euphorbiaceae affinities [30,31]. For the *trnQ–rps16* primers, we used (*trnQ*F-GCGTGGCCAAGYGGTAAGGC–*rps16*R-GTTGCTTTYTACCACATCGTTT), and for *petD* (D4) primers, we used (*petD*F—GGATTATGGGAGTGTRYGACTTG–*petD*RCTTTGTTATT GGGATAGGTGAA) from [32]. Polymerase chain reactions (PCR) were completed for the cpDNA regions using conditions reported in *Fontainea* studies by [12] and [13]. PCR products generated were sequenced by Macrogen Oceania (https://www.macrogen.com.au/). Chromatograms with trimmed primer ends for each gene fragment were imported into Geneious v11.1.4 [33] and trimmed based on quality. Gene fragments were individually aligned using MAFFT v7.308 [34] and concatenated in the following order: *trnQ–rps16*, *psbA–trnH*, and *petD* to produce a single sequence for each sample.

To obtain the three cpDNA regions of *A. moluccana* used for *Fontainea*, we performed a BLAST search of GenBank entries (nucleotide blast with algorithms "discontinuous megablast" or "blastn") using a reference *Fontainea* accession (FV017) for each cpDNA marker. Each of the three separate BLAST analyses resulted in well-resolved (100% coverage) alignments with GenBank regions: *petD*—HG971981.1:259–1070, partial *petD* gene, *trnQ*—MW322810.1:7865–8144, MW322810.1:7333–7725, *A. moluccanus* (sic) chloroplast, complete genome, *psbA*—MH837859.1:432–504, *A. moluccanus* isolate 1891, *psbA-trnH* intergenic spacer region, partial sequence; chloroplast. All BLAST results were concatenated in the same order as the *Fontainea* markers. Final alignments representing the seven *Fontainea* species and the additional outgroup sequences were processed in the R environment using the package DECIPHER [35]. We ran an initial alignment with all *Fontainea* accessions and the two outgroup accessions. Prior to constructing a phylogeny, we processed a refined alignment by running the "AdjustAlignment" function using an inbuilt algorithm, which systematically adjusts an alignment to resolve homologous regions. This algorithm provides a repeatable, fast function that moves gaps in an alignment left or right to find an optimal position defined as the position which increases the alignment score. As a result, low-quality local alignments are removed.

### 2.3. Reduced-Representation Genome Sequencing

Silica-dried leaf samples used in this study were sent to Diversity Arrays Technology Pty Ltd. (DArT, Canberra, Australia), where DNA extraction and the generation of DArTseq genome-wide markers using a proprietary NGS method [21,36] was conducted. Quality filtering of the resulting SNP data was implemented by an in-house designed R package "RRtools" v.1.0 [24] to remove poor quality SNPs that have a reproducibility score of <0.96, a call rate of $\geq 0.95$, missingness of <20%, and low-quality samples (high proportion of missing loci). Finally, to reduce the possible effect of linkage disequilibrium, SNPs were filtered to retain one SNP per locus for each sample. Thus, our filtering process retained 5608 loci and 17,122 SNPs, and no samples presented high missingness values to be omitted from downstream analyses. Sequences of the filtered DArTseq SNPs were converted into nexus format for alignment and phylogenetic analysis.

Due to the DArTseq platform only being optimised for *Fontainea-Baloghia,* we utilised a similar method to obtaining SNP sequences for *A. moluccana* as we performed for the cpDNA analysis. As expected, limited GenBank data were available for the nuclear regions of *A. moluccana*. Hence, we performed a BLAST search of GenBank entries (using the same algorithms as the cpDNA search) for nrDNA sequences of the first 10,000 from the filtered DArTseq dataset. Individual SNP sequences and a concatenated sequence

of the 10,000 SNPs were used in the BLAST searches. Results from the BLAST search yielded several retrotransposon and ITS regions (KU242993.1, KU242986.1, MH813127.1, MH813126.1), which were concatenated in the order listed. Final alignments for the DArTseq *Fontainea* phylogeny were performed as per the cpDNA construction method.

### 2.4. Phylogenetic Reconstruction

Phylogenetic reconstruction using the model-based maximum-likelihood (ML) approach was performed on the cpDNA concatenated alignment and DArTseq-adjusted alignment data separately to assess potential topology incongruence between the two datasets. For phylogenetic analysis, we used the R package, phangorn [37]. ML has proven accurate for testing phylogenetic hypotheses, particularly for large, genome-wide datasets for rapid, simple, and efficient use, which allows for selecting best-fit substitution models [38]. To evaluate clade support, substitution model selection was performed using the modelTest function to identify best-fit models for the cpDNA and DArTseq data under several model selection criteria, log-likelihood, Akaike information criteria (AIC), second-order AIC (AICc), and Bayesian information criteria (BIC). Both the cpDNA and DArTseq datasets were analysed with the concatenated sequences. To infer confidence values on the phylogenetic trees, we applied 1000 bootstrap replicates to the cpDNA dataset and 100 replicates for the DArTseq dataset, then plotted bootstrap support values on tree edges for the best-fit models. Lastly, an optimised, best-fit substitution model was run to allow for stochastic rearrangements. Optimised trees were then exported in Newick format for additional editing and visualised using FigTree v1.4.4 [39] using *A. moluccana* as the root node and compiled with vector editing software.

### 2.5. Estimates of Datasets Diverging to Different Topology

For the complimentary datasets from cpDNA markers and DArTseq SNPs, we assessed if the two marker types produced diverging estimates of phylogeny. Using the consensusNet function from the R package phangorn, we generated equal-length consensus network (CN) trees with a 0.3 proportional split to identify potential incongruence between the ML tree and CN of each dataset. Applying a CN tree to construct species phylogeny produces a tree that can display multiple incongruent evolutionary hypotheses [40]. This CN tree highlights conflicting species phylogenies as a result of introgression, stochastic processes, or sampling bias that other tree building methods, which violate assumptions from simple sequence-substitution models, are unable to perform [40].

### 2.6. Genetic Structure

Using the DArTseq SNP data, we examined the genetic structure between seven *Fontainea* species by running a principal component analysis (PCA). The PCA was performed in the R package, Adegenet [41], constructed from a pairwise Euclidian genetic distance matrix using the bitwise.dist function in the R package, Poppr [42]. In addition, to quantify levels of between-species genetic differentiation, we generated a pairwise fixation index ($F_{ST}$) matrix using the R package, DiveRsity [43].

## 3. Results

### 3.1. CpDNA Sequence Phylogeny

From 57 accessions of seven *Fontainea* species and two outgroup accessions, the aligned sequence length of the concatenated cpDNA markers was 2489 nucleotides. Phylogenetic trees based on the concatenated cpDNA sequences revealed differentiation between the seven *Fontainea* taxa, with varied levels of branch support. To present bootstrap values up to the terminal node level, we present ML trees with proportional, transformed branches and present the original ML trees available in the supplementary data (Figures S1 and S2).

Overall, there were three well-resolved lineages in the ML tree (Figure 2a), and these relationships were present in the consensus network (Figure 2b): Lineage 1 (L1) comprised *F. venosa* and *F. pancheri*, which formed sister species; lineage 2 (L2) is a distinct lineage

comprising *F. picrosperma*; the third lineage (L3) consisted of two sublineages, the first of which contained sister species of *F. rostrata* and *F. fugax* and the other being a mixed subclade of *F. australis* and *F. oraria* (Figure 2a). The GTR (General Time Reversible) + G (Gamma) + I (Invariant) model showed the best support of tree topology for the cpDNA (Table S3). Although lineage 1 had high bootstrap support (100), there was weaker support at the diverging nodes of L2 and L3.

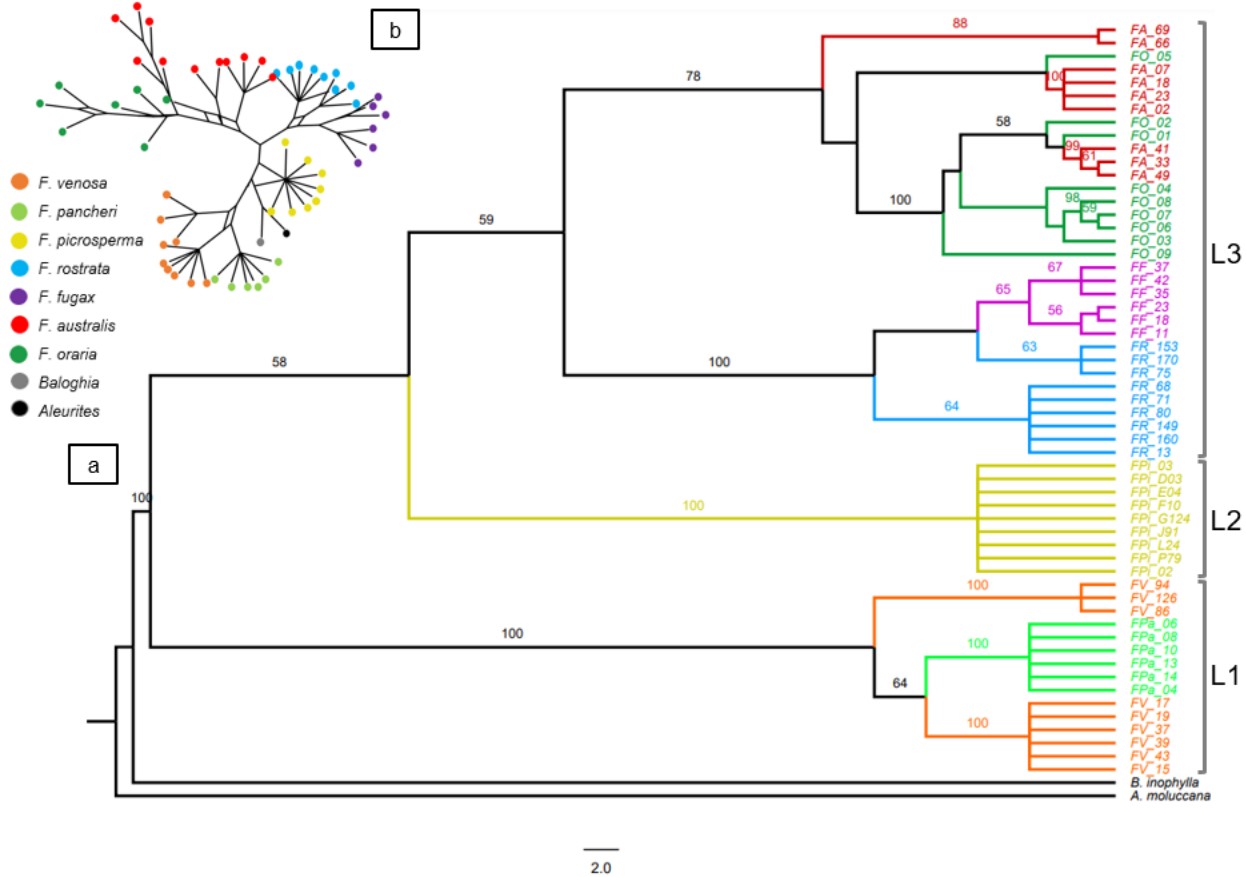

**Figure 2.** Phylogenetic tree with proportional terminal nodes (**a**) and consensus network (**b**) of the Austral-Pacific *Fontainea* complex constructed from individually aligned and concatenated cpDNA sequence markers using a maximum-likelihood analysis. *Baloghia inophylla* and *Aleurites moluccana* were used as outgroup representatives to root the tree. The values on the phylogenetic tree at the branches represent bootstrap support (>50) based on the best-fit GTR + G + I model, and the scale bar represents substitutions per site. The three *Fontainea* lineages recovered (L1, L2, L3) are represented outside the tip labels of the tree. Tree and consensus network tips are coloured by species; see inset legend.

　　*Fontainea venosa* and *F. pancheri* formed a sister lineage (L1) on the cpDNA ML tree and CN, with evidence of divergence between the geographically distant *F. venosa* populations, as shown by two sub-clade clusters in the CN tree (Figure 2a,b). One *F. venosa* subclade with the Boyne Valley accessions (FV86, FV94 and FV126) is strongly supported as a distinct subgroup. The remaining *F. venosa* accessions from the Gympie region formed a weakly supported sister group with *F. pancheri*.

　　*Fontainea picrosperma* was the only species to form a distinct, single lineage (L2) in the ML tree and CN. Accessions covered the geographical range of this species, which is only found in upland rainforest of the Wet Tropics region of northern Australia. *Fontainea fugax* and *F. rostrata* formed a very closely related, strongly supported sister group (Figure 2a). The close association among *F. fugax* and *F. rostrata* was also highlighted in the consensus

network tree (Figure 2b). There was weak branch support for the delineation of *F. fugax*, *F. rostrata*, *F. australis*, and *F. oraria* (L3) in the cpDNA ML tree (Figure 2a). However, further inspection of these species on the CN tree highlighted distinct clusters for each of these species (Figure 2b), which suggests that while branch support from the ML tree for species delineations was relatively weak, it is the most likely representation for the cpDNA sequence data.

Relationships among *F. australis* and *F. oraria* accessions were not clearly resolved on the cpDNA ML tree or network analysis (Figure 2a,b). Two *F. australis* accessions form a strongly supported sub-clade with the remaining *F. australis* accessions forming an additional subclade with some nested among *F. oraria* accessions. A network of boxes in the CN tree among *F. oraria* and most *F. australis* accessions suggest this may be a region of active diversification, which is an evolutionary hypothesis that requires further inspection. This likely reflects the geographical isolation and heterogeneous environment between *F. australis* populations, where accessions for this subclade were collected.

*3.2. DArTseq SNP Phylogeny*

The concatenation of all loci containing SNPs detected using DArTseq resulted in an alignment length of 33,170 nucleotides using the quality filtered data. Phylogenetic reconstruction using the DArTseq SNPs identified the GTR model as the best-fit model (Table S3), which produced strong values of support for the delineation of the major clades observed in the cpDNA phylogeny (Figure 3a).

Both the ML and CN tree representations of the DArTseq phylogeny recovered three main lineages, with different degrees of species-level to population-level resolution (Figure 3a). Generally, the DArTseq SNP data highlighted significant support at each species level and some congruent patterns with the cpDNA topology overall. Similarities with the cpDNA phylogeny highlighted the close genetic associations of *F. venosa* and *F. pancheri* (L1) and strong evidence for *F. picrosperma* as a distinct group (L2). *Fontainea rostrata* and *F. fugax* formed a sister subclade with *F. australis* and *F. oraria* (L3). Among the *F. rostrata*–*F. fugax* subclade, there was relatively weak support at the terminal nodes reflecting the geographic groups of these accessions. There was no presence of boxes in the network between most species (Figure 3b), which suggested a high level of congruence between the CN and ML phylogenetic trees. However, unlike the cpDNA data set, *F. oraria* and *F. australis* were clearly delineated from each other in the DArTseq ML tree and network analysis (Figure 3b).

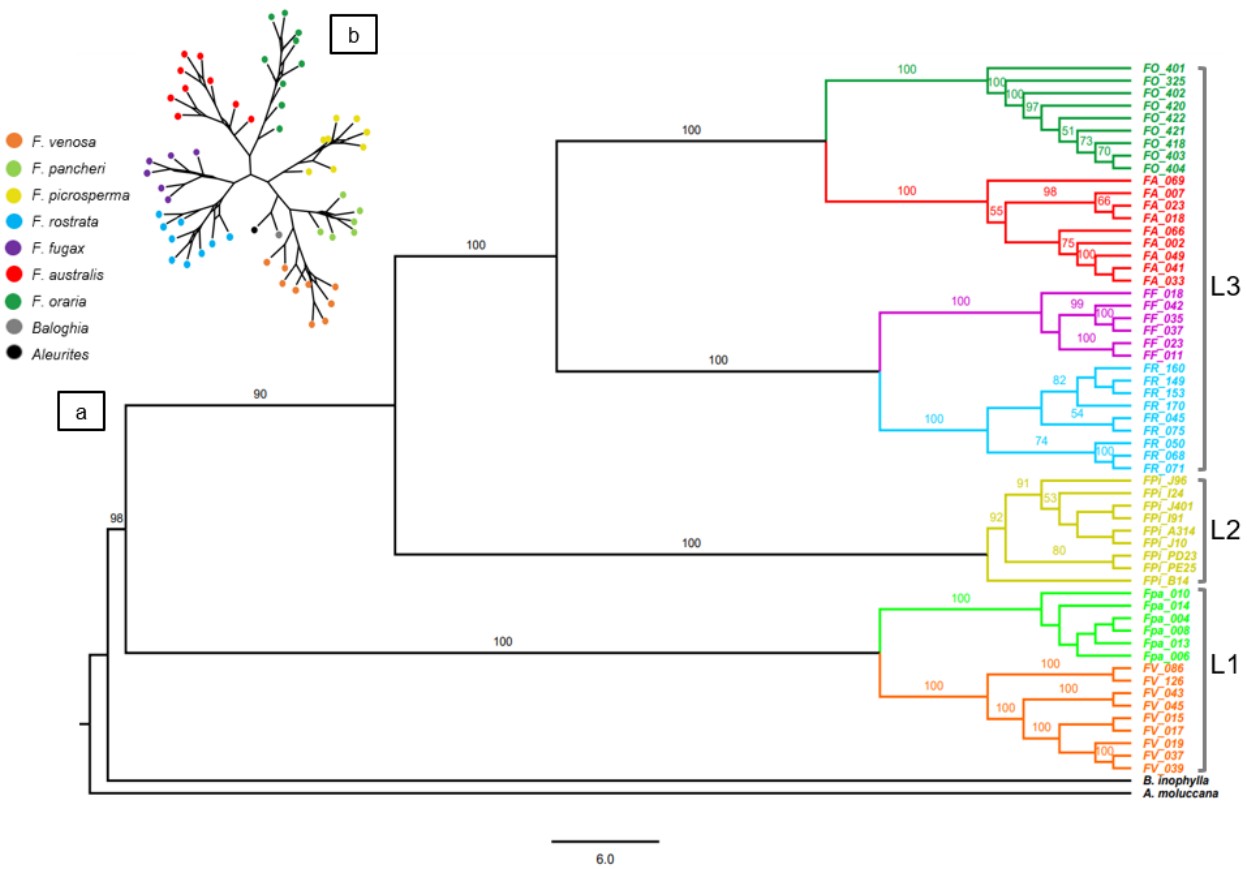

**Figure 3.** Phylogenetic tree with proportional terminal nodes (**a**) and consensus network (**b**) of the Austral-Pacific *Fontainea* complex constructed from the individually aligned and concatenated sequences of reduced-representation SNP markers (5608 loci) using a maximum-likelihood analysis. *Baloghia inophylla* and *Aleurites moluccana* were used as outgroup representatives to root the tree. The values on the phylogenetic tree at the branches represent bootstrap support (>50) based on the best-fit GTR model, and the scale bar represents substitutions per site. The three *Fontainea* lineages recovered (L1, L2, L3) are represented outside the tip labels of the tree. Tree and consensus network tips are coloured by species; see inset legend.

*3.3. Fontainea Population-Level Structure*

The patterns of phylogenetic associations present in the tree reconstructions was also reflected in the DArTseq analysis of genetic structure between *Fontainea* species. Three distinct clusters representing the major *Fontainea* lineages were observed in the PCA (Figure 4a), and these clusters support the relationships depicted in the phylogenetic trees constructed from both the cpDNA and DArTseq alignments. Furthermore, pairwise $F_{ST}$ values between species showed relatively high levels of genetic differentiation between the main clades (Table 2).

Moderate differentiation was indicated between the sister species within lineage 1, *F. venosa* and *F. pancheri* ($F_{ST}$ = 0.667), as well as lineage 3, *F. fugax* and *F. rostrata* ($F_{ST}$ = 0.539). There was relatively weak differentiation observed between the southern taxa of *F. australis* and *F. oraria* $F_{ST}$ = 0.294 and close clustering on the PCA. These patterns were in agreement with the levels of support and clustering of branches in both the ML phylogenetic trees and network analysis for these two species for these two species.

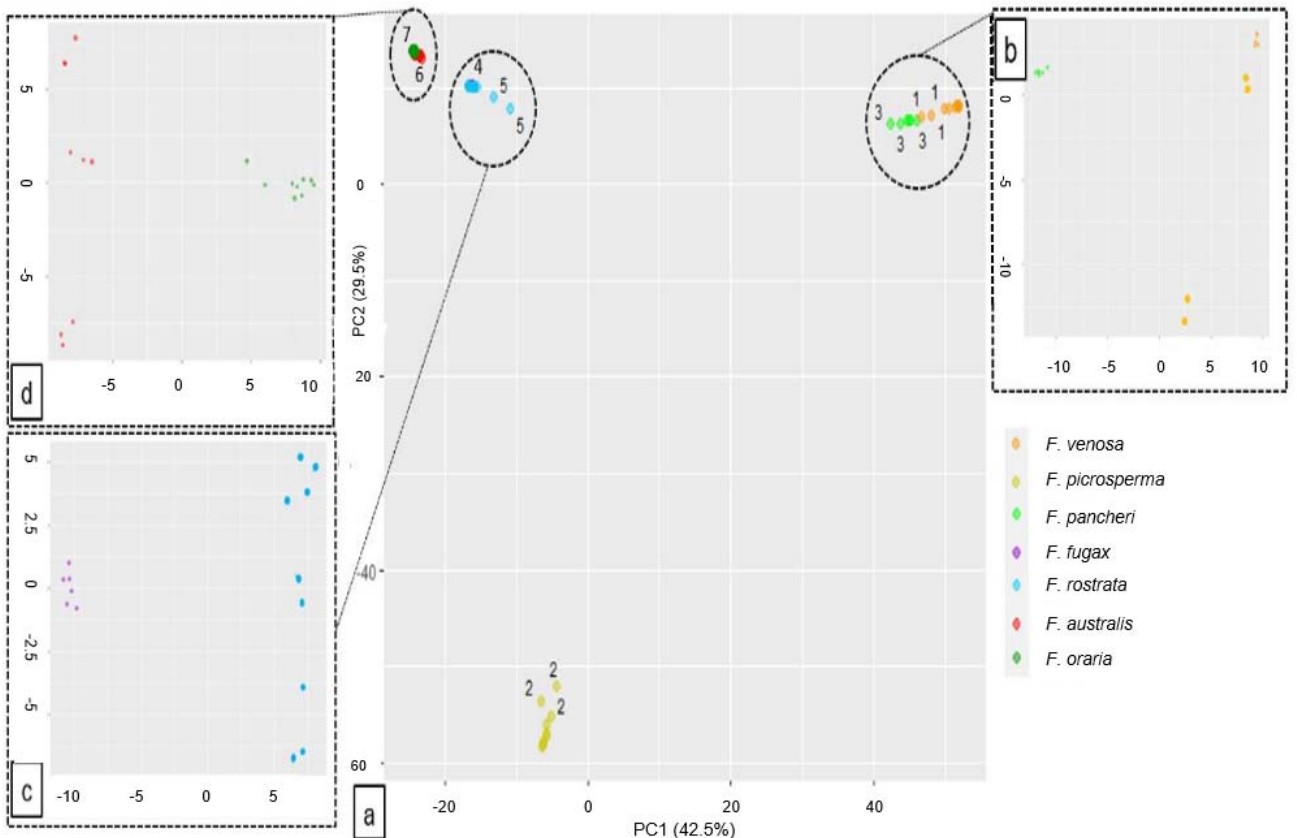

**Figure 4.** Principal component analysis (PCA) of the Fontainea genus from reduced-representation DArTseq SNP data (**a**) showing genetic grouping along axes 1 and 2. Inset PCAs represent the three putative genetic groups of sister species ((**b**) *F. venosa-F. pancheri*, (**c**) *F. fugax-F. rostrata*, (**d**) *Fontainea australis-F. oraria*) identified from phylogenetic analyses, which are magnified for clearer resolution of distribution of the closely grouped species along each respective axes.

**Table 2.** Pairwise $F_{ST}$ matrix of the Austral-Pacific *Fontainea* complex from reduced-representation SNP data (5608 loci). Coloured boxes represent gradient level of genetic differentiation among each species. Gradient scale is indicated below table.

| | *F. venosa* | *F. pancheri* | *F. picrosperma* | *F. fugax* | *F. rostrata* | *F. oraria* | *F. australis* |
|---|---|---|---|---|---|---|---|
| *F. venosa* | - | 0.667 | 0.97 | 0.965 | 0.963 | 0.962 | 0.963 |
| *F. pancheri* | 0.667 | - | 0.981 | 0.976 | 0.975 | 0.971 | 0.972 |
| *F. picrosperma* | 0.97 | 0.981 | - | 0.965 | 0.961 | 0.953 | 0.954 |
| *F. fugax* | 0.965 | 0.976 | 0.965 | - | 0.539 | 0.863 | 0.865 |
| *F. rostrata* | 0.963 | 0.975 | 0.961 | 0.539 | - | 0.861 | 0.861 |
| *F. oraria* | 0.962 | 0.971 | 0.953 | 0.863 | 0.861 | - | 0.294 |
| *F. australis* | 0.963 | 0.972 | 0.954 | 0.865 | 0.861 | 0.294 | - |

0 ▬▬▬ 1

A closer inspection of the PCA highlighted some evidence of genetic partitioning within *Fontainea* sister species (Figure 4). *Fontainea pancheri* was densely clustered, while *F. venosa* was distributed in several clusters along the second principal component (Figure 4b). *Fontainea fugax* and *F. rostrata* showed similar patterns of distribution on the zoomed PCA (Figure 4c). *Fontainea fugax* accessions was relatively tightly grouped but distinct from *F. rostrata*, which was distributed continuously along the second principal component. The *F. australis–F. oraria* subclade showed some subtle deviation from this pattern (Figure 4d). *Fontainea australis* are distributed in several distinct groups along the second principal component though not completely discrete from accessions of *F. oraria*. *Fontainea oraria*

was distributed across the first principal component with some accessions in proximity to *F. australis*.

Genetic differentiation between species did not always indicate geographic partitioning among all species. As an example, *F. venosa* is closely aligned with the isolated Pacific Island accessions of *F. pancheri*. Conversely, relatively high levels of genetic differentiation were observed from the geographical neighbour accessions of *F. rostrata* and *F. venosa*.

## 4. Discussion

Our phylogenetic analyses of cpDNA sequence markers and DArTseq SNP datasets highlighted clear genetic partitioning within and between the Austral-Pacific *Fontainea* complex. Each respective dataset identified three major lineages of the *Fontainea* genus, some of which comprised sister species, largely supported by their geographic distribution, genetic structure analyses and previous taxonomical treatments. Within the subclade that included the southern species, *F. australis* and *F. oraria*, we observed genetic signatures suggesting geographic partitioning with partial sharing of plastid genotypes between the species. A distinct genetic signal was observed for *F. picrosperma* from tropical northern Australia. Significantly, evidence from molecular analysis suggests the *F. venosa–F. pancheri* subclade is the putative, earliest diverging lineage of the genus. A fossil *Fontainea* affinity from the Oligocene-Miocene period could arguably be placed at a stem position of divergence from modern crown *Fontainea*. If this hypothesis is found to be true, ancestors of the extant *Fontainea* species have likely survived for millions of years by tracking suitable biome shifts, adaptation to complex landscapes, and transoceanic dispersal.

All phylogenetic representations recovered *F. venosa* and *F. pancheri* in a sister species lineage at the earliest diverging position within the genus and highlighted a distinct geographical group of *F. venosa* at the northern range limit of the species. The basal position of *F. venosa* nested with *F. pancheri* is particularly insightful, as *F. venosa* is endemic to Australia, whilst *F. pancheri* occurs in New Caledonia and Vanuatu, some 1400 km east in a region of the Pacific, which shares a number of primitive lineages of Gondwanan origin with Australia [44,45]. Many studies agree that land connections between eastern Australia and the Zeelandia plate (part of which formed the present-day New Caledonia) were possible up to 65 Ma [22,23,46]. This suggests divergence between the Australian and New Caledonian *Fontainea* would have to be dated at least 65 Ma to be vicariant as a consequence of continental drift. If the *Fontainea* fossil affinity, *Fontainocarpa*, dated from the Oligocene-Miocene period (20–32 Ma) [2] proves to be, as we suggest, at a stem position to modern crown taxa, our phylogenetic analysis indicates the arrival of *Fontainea* in New Caledonia is too young to be explained by Gondwana vicariance. Furthermore, several studies suggest that the New Caledonia archipelago was periodically submerged before the late Eocene, around 56–40 Ma [47–49]. Crisp and Cook [23] presented divergence time estimates between a number of sister plant taxa from Australia and New Caledonia, finding that the oldest divergence date was around 42 Ma. Transoceanic dispersal via ocean current or animal vectors during this period is plausible, and a scenario was found to explain numerous dispersal and colonisation events of the Sandalwood genus (*Santalum* L.) throughout the Pacific from eastern Australia [50] and a lineage of Araucaria during the Eocene [48]. In addition, a late Miocene arrival in New Caledonia of a *Psychotria* clade, which originated from Australia [51], provides support to a long-distance dispersal theory. Hence, we propose that the divergence and colonisation of *F. pancheri* in New Caledonia was likely a result of genetic drift following a long-distance dispersal event founded from Australian *F. venosa*, or they are the same, geographically disjunct taxon with some degree of intra-specific variation. *Fontainea pancheri* and *F. venosa* are almost morphologically identical except for their fruit features and may be conspecific. New Caledonia types of *F. pancheri* have a ridged endocarp with numerous, scattered foramina compared to the obtusely or subglobular ridged endocarp with few foramina of *F. venosa* [3]. Furthermore, variation (inflorescence and endocarp structure) among *Fontainea* collections, including the

syntypes of *F. pancheri*, indicates that additional, poorly known, undescribed species may be confused with the taxon [3].

*Fontainea picrosperma* from the upland Wet Tropics of northern Australia was highlighted as a distinct lineage with no close genetic associations to other *Fontainea*. This is in agreement with the geographic isolation of *F. picrosperma*, which is found ~1500−1800 km north of the southeast QLD and northern NSW *Fontainea* species. This species is cultivated for its unique phytochemical properties [3] and is also phenotypically distinct, with appressed, pilose leaves along the midvein and growing to a height of 25 m [3]. In addition, the *F. picrosperma* endocarp typically has four to five ridges [3]. *Fontainea picrosperma* was the only species that showed strong genetic differentiation from all other congeners, which may signal a long period of vicariance coupled with distinct selection pressures. A variety of interacting factors have likely afforded refugia for *F. picrosperma*. Over the last 30 Ma, the drying climate, growing fire frequency, limited topographical relief, and edaphic features have largely influenced the present day distribution of rainforest vegetation [44] ]. The more recent extreme inter-glacial cycles during the Pleistocene, ~5 Ma, have also significantly shaped the expansion and contraction dynamics of rainforest species along eastern Australia [12,44]. Noted for a composition of low phylogenetic diversity compared to other areas in the Australian Wet Tropical biome, this upland region has retained its Gondwanan character as a result of the high relief-ameliorating paleoclimate fluctuations [52]. Based on morphological evidence [3] and ancient signals of biotic exchange from the contact of the Sahul and Sunda continental plates [53], we may infer that *F. picrosperma* could potentially form a sister subclade with the two PNG species, *F. borealis* and *F. subpapuana*. However, we must be conservative in this assumption, as we have also presented evidence of genetic differentiation between the geographical neighbour species *F. venosa* and *F. rostrata*.

We found a sister relationship among *F. rostrata* and *F. fugax*, which are geographically isolated (>150 km) but share many phenotypic similarities in leaf dimensions, with the presence or absence of a swollen petiole and reproductive distinctions used for taxonomical delineation. Male flowers of *F. rostrata* typically have 28–40 stamens compared to 24 for *F. fugax* [4]. The genetic relationship among *F. fugax* and *F. rostrata* may represent a signal of expansion from previously bottlenecked populations across a broader region into suitable conditions. This pattern of expansion and contraction following paleoclimatic cycles may in fact typify the evolution of *Fontainea* due to the isolated, geographically disjunct character of the genus. More broadly, the contraction–expansion cycles of Australian rainforest flora across deep time are well-documented [54–56]. Despite some populations of *F. venosa* and *F. rostrata* being close regional neighbours, strong genetic differentiation indicated long-term reproductive isolation between these species. Conroy, Shimizu-Kimura [13] reported low levels of genetic diversity and evidence of genetic drift within northern populations of *F. rostrata*. These factors can indicate range expansion within a species [57] and may provide contemporary evidence of the expansion of *Fontainea* when suitable habitat once existed across much of eastern Australia.

Our phylogenetic reconstructions highlighted shared plastid genotypes between *F. australis* and *F. oraria* the southernmost distributed species of the genus, which occurs in littoral, coastal rainforest in New South Wales, together with evidence of genetic divergence between some geographical neighbour populations of *F. australis*. Specifically, *F. australis* accessions from the northern range appear to be genetically divergent from other *F. australis* populations sampled for this study. These divergence signals were not clear in the DArTseq phylogeny. It is possible our analyses identified accessions within the southern subclade at an early stage of divergence, which may signal a genetic cline or adaptation to the diverse environments within the region sampled for these species. Patterns of genetic differentiation that contributed to species differentiation driven by climatic and abiotic differences have been highlighted in a number of plant species and assemblages along eastern Australia [58–60]. This unresolved genetic signal may be improved by further sampling across the species range as well as further surveying for new populations, together with the investigation of the influence of environmental factors to better define how this

species is positioned within the phylogeny of the genus. Crucially, there are a number of *F. australis* populations yet to be verified from historical records that were not sampled at the time of this study and are in closer proximity to *F. oraria*. Together, these key findings may have significant taxonomical and conservation implications, as *F. oraria* is classified as critically endangered [61], and *F. australis* as vulnerable [62]. The current delineation between *F. australis* and *F. oraria* is based on variations between basilaminar gland distance, female floral axes length, and endocarp features [20]. For example, basilaminar glands in *F. australis* are between 5–22 mm above the base of the laminar, whereas in *F. oraria*, they are noted to be much closer at 0.5–4 mm above the laminar base [3]. However, taxonomical classifications founded from morphological features have been improved for numerous plant species when genetic analysis has been included in subsequent revisions [50,62]. Therefore, a genetic and morphological examination of the southern *Fontainea* subclade, which represents the complete geographical distribution, is suggested as a priority for the contemporary classification and conservation management of this group.

Additionally, our phylogenetic analyses showed the *F. australis–F. oraria* subclade is sister to the *F. fugax–F. rostrata* subclade. This supports early research by Rossetto, McNally [17], which suggested *F. australis* was founded from *F. rostrata* and may indicate that *F. rostrata* was once more widely distributed. We suggest the phylogenetic signal for the southern subclade represents a history of evolution in response to the climate oscillations of the Quaternary. Climate fluctuations during this period led to contraction and expansion cycles of rainforest, which was particularly intense in subtropical regions [63]. *Fontainea australis* likely evolved in a historically more contiguous habitat, which followed a period of major volcanic activity (~23–25 Ma) and formation of the relief of the Wollumbin Caldera [64]. The unique topographical relief of the caldera may have provided refugial areas during paleoclimate-induced loss of habitat and could have had a major effect on the genetic differentiation within the *F. australis–F. oraria* subclade. Evidence of shared plastid genotypes indicated that there may also be a hybrid zone from secondary contact between these sister species. We can infer a scenario where *F. australis* and *F. oraria* evolved in spatially segregated habitats, with selection from a range of environmental factors driving divergence between the two species and a zone of hybridization between differently adapted types. However, since European settlement, the habitat of these two species has been intensely cleared, with <1% of its original area remaining [65]. Therefore, unless new populations of *F. australis* and/or *F. oraria* are discovered, a clear signal, which highlights an area of contact and divergence between these species, and ultimately an accurate model of speciation may remain elusive. It is also likely that dispersal limitations have played a key role in genetic structuring between *F. australis* populations.

Now-extinct megafauna were once central to the dispersal of fleshy-fruited subtropical rainforest species [66]. However, given the combination of low genetic diversity, a lack of dispersal observations, and the clumped and fragmented distribution seen in extant species of *Fontainea*, gravity and hydrochory have been proposed as the contemporary mechanisms of diaspore transport for the genus [12]. Nevertheless, there has been no study so far of the dispersal syndromes of any *Fontainea* species; yet, the genus produces a fleshy, indehiscent fruit, which is relatively rare in Euphorbiaceae [8]. When considering that these fruit features are unusual across Euphorbiaceae, it suggests an evolutionary shift from mechanical to animal-mediated dispersal [8]. Although *Fontainea* fruit contain a relatively large, woody endocarp, all species have fruit less than 30 mm (length and width), which is within the size range that can be ingested by several large extant, Australian avian dispersers [67]. Therefore, it is not possible to equivocally discount the role animal vectors may play in present-day *Fontainea* distribution and geneflow. This is a key area of the ecology of the genus that would benefit from further attention. However, pollen dispersal of *F. picrosperma* has a limited range of 30 m [16], and we therefore anticipate that similar pollen-flow dynamics between other isolated *Fontainea* populations could limit gene flow and increase genetic structuring and drift.

Finally, our phylogenetic reconstructions have presented valuable knowledge of the genetic relationships of *Fontainea* and have also raised some important considerations regarding the taxonomical hierarchy between some species. Generally, the phylogenetic position of the seven *Fontainea* species among both datasets follows the taxonomic key outlined in Jessup and Guymer [3], which delineated the earliest diverging species (*F. pancheri* and *F. venosa*) with the presence of glabrous (without hairs) ovaries in crown species (*F. australis*, *F. oraria*, *F. picrosperma*, *F. rostrata*) exhibiting villous (long soft hairs) ovaries. *Fontainea fugax* was first described by Forster [4] as closely allied to *F. rostrata*, with an original collection being referred to as this species [5]. Our phylogeny indicated *F. fugax* and *F. rostrata* as sister species and also supports the taxonomic distinctions of Forster [4]. We also recovered *F. australis* and sister to *F. oraria*, which was expected with the close taxonomical affinities [3]. Yet, we also identified geographically and genetically distinct variation within species (*F. venosa* and *F. australis*), which may require further attention. Given the conservation significance of the genus, future research would benefit from an integrated examination of the genetic mechanisms, morphology, and environmental influence of divergence between *Fontainea* species to clearly resolve the taxonomy of the genus.

## 5. Conclusions

The fragmented, transoceanic distribution and limited fossil evidence make *Fontainea* an excellent model for investigating southern hemisphere processes of species divergence from rainforest plant lineages. Our phylogenetic construction and additional molecular analysis have provided a framework that can be used to form preliminary inferences about the historical biogeography and evolution of *Fontainea.* We suggest that a complex series of long-distance dispersal, in tandem with vicariance as a result of plate tectonics and paleoclimatic history, have led to the emergence of seven species within *Fontainea*. Our data indicate three major *Fontainea* lineages. However, clear delineation between all species is not resolved. There is some uncertainty of the genetic relationships between the sister species, particularly the southern subclade of *F. australis* and *F. oraria*. A detailed study that combines genetic, morphological, and environmental data would likely further clarify the taxonomical hierarchy. Further, the reduced-representation data were able to provide robust species delineation, which supports this method as a valuable tool in phylogenetic research. Our data also suggest that *F. venosa*–*F. pancheri* is the earliest surviving lineage currently distributed in central and eastern Australia. Several radiation events, including a cross-ocean dispersal, could have propelled the divergence among the *Fontainea* species spanning millions of years to the most contemporary clade of *F. australis* and *F. oraria*. Overall, this study advanced our knowledge of the phylogenetic and evolutionary history of the scientifically and commercially significant rainforest genus, *Fontainea*, and its survival through to the present time.

**Supplementary Materials:** The following supporting information can be downloaded at: https://www.mdpi.com/article/10.3390/d14090725/s1, Figure S1: Maximum-likelihood phylogenetic tree of the Austral-Pacific Fontainea complex constructed from individually aligned and concatenated cpDNA sequence markers based on the best-fit GTR + G + I model. *Baloghia inophylla* and *Aleurites moluccana* were used as outgroup representatives to root the tree. The scale bar represents substitutions per site and tip labels are coloured by species; see inset legend; Figure S2: Maximum-likelihood phylogenetic tree of the Austral-Pacific *Fontainea* complex constructed from the individually aligned and concatenated sequences of reduced-representation SNP markers (5608 loci) based on the best-fit GTR model. *Baloghia inophylla* and *Aleurites moluccana* were used as outgroup representatives to root the tree. The scale bar represents substitutions per site and tip labels are coloured by species; see inset legend; Table S1: Values for nucleotide substitution models used to select best-fit model of *Fontainea* phylogenetic tree construction from concatenation of three cpDNA markers; Table S2: Values for nucleotide substitution models used to select best-fit model of *Fontainea* phylogenetic tree construction from concatenation of DArTseq SNPs (33170 nucleotides); Table S3: Accession details

of seven *Fontainea* taxa with indication of markers assignments and sampling locations (accession numbers in brackets represent DArTseq labels of *F. oraria*).

**Author Contributions:** Conceptualization A.J.B., R.W.L., G.C.C., P.W.R. and S.M.O.; methodology, A.J.B., R.W.L., G.C.C., S.Y., M.R., A.T-B. and S.M.O.; software, A.J.B., R.W.L., G.C.C., S.Y. and A.T-B.; formal analysis, A.J.B., R.W.L., G.C.C., S.Y., M.R., A.T.-B. and S.M.O.; writing—original draft preparation, A.J.B.; writing—review and editing, A.J.B., R.W.L., G.C.C., S.Y., M.R., A.T.-B., L.M. and S.M.O.; supervision, G.C.C., M.R. and S.M.O.; project administration, G.C.C. and S.M.O.; funding acquisition, P.W.R. and S.M.O. All authors have read and agreed to the published version of the manuscript.

**Funding:** A.J.B. is funded by an Australian Government Research Training Program Scholarship and a 2020 Rotary-USC Community Fund Scholarship. This work was supported by EcoBiotics Ltd. and the University of the Sunshine Coast.

**Institutional Review Board Statement:** Not applicable.

**Data Availability Statement:** Data available upon request.

**Acknowledgments:** A.B. wishes to thank E.A.B and F.J.B. for office space and meals during CV19 lock-down periods.

**Conflicts of Interest:** S.M.O. is a director and shareholder of QBiotics Group Ltd. P.W.R. is a director, employee, and shareholder of EcoBiotics Ltd. and QBiotics Group Ltd. The remaining authors declare no conflict of interest.

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
