# Peer review of "Phylogenetic Reconstruction of the Rainforest Lineage Fontainea Heckel (Euphorbiaceae) Based on Chloroplast DNA Sequences and Reduced-Representation SNP Markers"

_diversity, doi:10.3390/d14090725_

Round 1

Reviewer 1 Report

Dear Author and Editor

The article presents the phylogenetic reconstruction of the rainforest lineage Fontainea Heckel Euphorbiaceae) based on chloroplast DNA sequences and reduced-representation SNP markers is very attractive to the reader.

Although the research object is limited in distribution in Australia, it has a clear significance in the study of plant taxonomy.

Especially using the reduced-representation SNP markers technique to compare with chloroplast DNA sequences. Therefore, the results are highly reliable.

The study is thorough, systematic and very well written.

Author Response

Reviewer 1 did not raise any specific comments that were required to be addressed.

Reviewer 2 Report

This manuscript assessed the phylogeny of seven Fontainea taxa from the Australian and Pacific Island complex using chloroplast DNA sequence data and reduced-representation genome sequencing. The result advanced knowledge of the phylogenetic and evolutionary history of the rainforest genus, Fontainea. In general, the manuscript is clearly written in professional, unambiguous language, and the manuscript is interesting for the general reader of Diversity. I recommend accept the manuscript after a minor revision.

However, there are some questions which need to address before acceptance in the Journal. 

1. The content of Abstract should be structured better again.

2. The Figures 1 and 4 are not clear, the clear version should be changed.

3. The information of 7 species and 59 accessions should be described detailedly in the table 1 (age, Longitude, Latitude and altitude et al.).

4. The content of combination of the molecular result in the manuscript and the previous morphological, phytochemical and anatomygical information should be added in the part of dicussion.

In conclusion, I think that the manuscript should be revised in detail before accepted.

Author Response

Please find responses to the four key points from Reviewer 2 addressed below.

1 - Thank you for this valuable feedback. We have edited the abstract to improve the flow and structure (lines 22-32). 

2 - Thank you for highlighting this point. Figure 1 and Figure 4 were noted as not clear. We anticipate this is the resolution of each image and have therefore used an image editor to improve the resolution of both figure. We have also improved the text and number which support Figure 4.  

3 - This is a well-considered comment, which we appreciate. Where possible we have revised the relevant table in the supplementary data to include location data as latitude, longitude. However, due to some species being only found on private land, and the rare nature of several species, some location points have been deliberately inflated to protect the plants from poaching or damage and to protect the privacy and safety of private landholders.  Given the wide geographic range of the genera, we believe this approach gives appropriate context for the locations. We have included approximate elevation for each species in the Table 1 information as an additional column and made a brief note in the methods leaf material was collected from mature individuals (line 182 and Table S1).

4 - 

This is a valuable comment. Some additional points relating to the phylogenetic results and morphological relationships/distinctions among species have been included in the discussion. We agree, a comparison of the phytochemical properties of Fontainea may add value to the discussion. However, to date, there is only phytochemical information available for Fontainea picrosperma and a thorough discussion of this subject would not provide additional weight to the study without the context of more Fontainea species. We also suggest this is outside the scope of the study but would be a well-regarded aspect which could be pursued at a later time (lines 420-545).

Reviewer 3 Report

This is a very interesting and original work using an innovative approach to solving phylogenetic relationships and Phylogenetic reconstruction of the rainforest lineage Fontainea

Heckel (Euphorbiaceae). The work is at a high scientific and editorial level.

I only have two observations. I hope the authors will refer to them.

Abstract

In my opinion, the sentence: "One species

is cultivated commercially as the source of a cancer therapeutic and despite the threatened status of

many constituent taxa, the phylogenetic relationships of the genus have not been explored. "

The mere fact of being an endangered species should not be used as an argument to analyze phylogenetic relationships automatically. The threat does not always justify or explain this type of research. This is not always the case.

Editorial matters:

to correct: keyword1 and probably the numbers are not needed

M&M:

I don't really understand the reduction. If I understood correctly, there were 10,000 raw reads, then 33,170 nucleotides were left (5,608 loci?). Please logically explain the path of data analysis and reasoning and add a description in the manuscript in the appropriate section.

I also don't really understand the data analysis path. 10.000 raw reads to BLAST? I am asking for a more detailed description of the procedure.

Author Response

Please see responses to Reviewer 3 comments below.

Abstract: This is an excellent point. We understand the conservation status may not initially seem to support a phylogenetic investigation. However, when considering the effort and resources provided to threatened species, an initial step in conservation genetics is resolving any potential taxonomic uncertainties. For example, a species considered geographically widespread and abundant, may in fact contain a complex of distinct or cryptic taxa which prove to be rare (Franklin et al. 2009). In addition, taxonomically distinct species may in reality be genetically aligned with a close congener and result in taxonomical revision and improved conservation management advice.  In the case of our study species, it is highlighted in our introduction that Rossetto et al (2000) highlighted that two of the threatened (one critically endangered) species included in our study have taxonomic uncertainty, which reinforces the need for phylogenetic clarity from both a commercial and ecological/conservation standpoint: “Rossetto et al., (2000) highlighted some populations of F. australis were genetically different from other populations and did not cluster with morphologically similar F. oraria. Clearly, a molecular-based study that examines relationships within and between Fontainea species is needed to better manage conservation efforts for this commercially and ecologically significant rainforest genus”. In addition, the conservation advice for several Fontainea species specifically highlights the need for phylogenetic analysis. We believe in this case, this justifies the statement of intent in the abstract.

Editorial matters: Keywords section has been edited as suggested.

Final comment - M & M: This is a key point and we agree that this could have been explained better, therefore we have edited for clarity in the manuscript. Briefly, we have expanded this section in the methods to explain the number of nucleotides and loci retained are from the DArTseq filtering process. The additional alignment of Aleurites sequences is from several markers, however the maximum length of sequences retained from the filtered DArTseq dataset is what is quoted. To conduct the BLAST search, and identify Aleurites SNP markers which align with the Fontainea DArTseq markers we used filtered dataset. However, as expected, we only found a limited amount of markers with similar sequences (lines 224-234).